# Evaluation of cytological diagnostic accuracy for canine splenic neoplasms: An investigation in 78 cases using STARD guidelines

**Marco Tecilla**⊗, **Matteo Gambini**[ORCID]⊗*, **Annalisa Forlani**¤, **Mario Caniatti, Gabriele Ghisleni, Paola Roccabianca**

Diagnostic Pathology Service, Dipartimento di Medicina Veterinaria (DIMEVET), Università degli Studi di Milano, Lodi, Italy

⊗ These authors contributed equally to this work.
¤ Current address: IDEXX Laboratories, Wetherby, West Yorkshire, United Kingdom
* matteo.gambini@unimi.it

**Data Availability Statement:** All relevant data are within the manuscript and its Supporting Information files.

## Abstract

Cytology represents a useful diagnostic tool in the preliminary clinical approach to canine splenic lesions, and may prevent unnecessary splenectomy. However, few studies have evaluated diagnostic accuracy of cytology in the diagnosis of canine splenic neoplasms. The aim of this study was to determine overall accuracy, sensitivity, specificity, positive and negative predictive values (i.e. diagnostic accuracy indexes) of cytology for canine splenic neoplasms following Standards for the Reporting of Diagnostic Accuracy Studies (STARD) guidelines. A consecutive series of canine splenic cytological samples was retrospectively retrieved from the database of the Diagnostic Pathology Service of the Department of Veterinary Medicine (DIMEVET—University of Milan). Histopathology was set as the diagnostic reference standard. Cytological cases were enrolled when slides were available for review and when the same lesion was submitted for histopathology. Seventy-eight (78) lesions were included in the study. By histopathology, 56 were neoplastic and 22 were non-neoplastic. Cytology had an overall accuracy of 73.08% (95% C.I. 61.84%-82.50%), sensitivity of 64.29% (95% C.I. 50.36%-76.64%), specificity of 95.45% (95% C.I. 77.16%-99.88%), and positive and negative predictive values of 97.3% (95% C.I. 84.01%-99.60%) and 51.22% (95% C.I. 42.21%-60.15%), respectively. Low sensitivity and negative predictive value were balanced by very high specificity and positive predictive value. When positive for neoplasia, cytology represents a useful diagnostic tool to rule in splenic neoplasia, prompting surgery independently from other diagnostic tests. Conversely, a negative cytological result requires additional investigations to confirm the dog to be disease free.

## Introduction

Ultrasonographic examination of nodular splenic lesions in dogs is not reliable to differentiate with certainty benign and malignant processes, necessitating the use of additional, ideally minimally invasive, diagnostic tests [1]. Hemangiosarcoma (HES) is the most common primary

**Funding:** The authors received no specific funding for this work. All the authors were employed by the Università degli Studi di Milano at the time by which the study was conducted and concluded. One of the authors [A.F.] is currently employed by IDEXX Laboratories (Wetherby, West Yorkshire, UK). This Company did not provide support in the form of salary for any of the authors at the time by which the study was conducted, neither had any additional role in the study design, research material supply, data collection and analysis, decision to publish, or preparation of the manuscript.

**Competing interests:** One of the authors [A.F.] is currently employed by IDEXX Laboratories (Wetherby, West Yorkshire, UK) which did not provide support in the form of salary for any of the authors at the time by which the study was conducted, neither had any additional role in the study design, research material supply, data collection and analysis, decision to publish, or preparation of the manuscript. This does not alter our adherence to PLOS ONE policies on sharing data and materials.

splenic malignant tumor of dogs [2]. Still, HES represents fewer than 25% of overall splenic lesions [3] with up to 74% of dogs being diagnosed with benign lesions [4] such as hematoma and hyperplasia [2,5]. However, due to the poor prognosis associated with HES, splenectomy is still the routine approach to most canine splenic masses for both diagnostic and therapeutic purposes [6–8].

While the two-month post-splenectomy survival rate is lower in dogs with HES (32%) compared to dogs with hematomas (83%), the general survival rate after splenectomy is 52%, regardless of underlying splenic pathology [3]. While this can be interpreted as fair survival data, a proportion of dogs (7.6%) will develop complications secondary to splenectomy because of thrombotic or coagulopathic syndromes [9]. Additional adverse effects following splenectomy in dogs have included peri- and post-operative ventricular arrhythmias [9–11], reduced blood filtration and renewal [12,13], impairment of humoral immune response [14], reduced immune-surveillance against bacteria and parasites [15–19], and higher incidence of gastric dilatation-volvulus [6,20–22]. For these reasons, any preoperative diagnostic approach to splenic lesions, including cytology, may be beneficial in preventing unnecessary splenectomy. Notwithstanding the common belief that splenic aspiration can be dangerous especially when investigating cavitated masses [6,7,23], complications from splenic aspiration procedures are rarely elicited even in thrombocytopenic animals [8,24–26]. For comparison, in human medicine splenic fine needle aspiration (FNA) cytology is seldom associated with complications [27,28], resulting in 5.2% secondary complications with fewer than 1% considered severe and consisting mostly of controllable hemorrhage [29].

Thus, attempts to minimize unnecessary splenectomy should prompt an increased use of additional diagnostic techniques as preoperative screening tests to characterize splenic disease. Fine needle aspiration cytology can provide diagnostic information useful to distinguish inflammatory, benign and malignant nodular lesions and to assess generalized splenomegaly [7,24,30].

Despite the relatively high frequency of splenic diseases in dogs, data regarding usefulness and validity of diagnostic cytology are fragmentary. In veterinary medicine, no studies have comprehensively assessed overall accuracy, sensitivity, specificity, positive and negative predictive values of canine splenic cytology against histopathology utilizing the Standards for the Reporting of Diagnostic Accuracy Studies (STARD) guidelines [8,23–25,31–33]. STARD guidelines have been created to avoid incomplete reporting in diagnostic accuracy studies and to improve the general quality of the latter, reducing problems related to study identification, critical appraisal, and replication [34]. Overall agreement between cytology and histology of canine splenic lesions is the most frequent index reported, ranging from 38 to 100% [8,23–25,31–33]. Specifically, this index has been evaluated on a limited number of splenic cytological specimens (range 5–40) [8,23–25,31–33]. In most reports, sensitivity, specificity, positive and negative predictive values of splenic cytology are not calculated and cannot be properly estimated [8,23–25,31–33] because caseloads simultaneously include multiple species [24,25,31–33], multiple tissues and organs [31–33], or "equivocal" or "provisional" cytological and histological diagnoses [23,24,33]. Application of STARD guidelines in the current study allowed cross-tabulation of cytological results (i.e. the index test) against those of histopathology (i.e. the reference standard) to generate sensitivity and specificity data [34]. These data have not been included in previous studies and will be useful for future researchers comparing diagnostic methods for canine splenic neoplasms.

To avoid unnecessary splenectomy, a diagnostic test with a high sensitivity and negative predictive value is desirable because these indexes measure the percentage of diseased dogs correctly diagnosed with splenic neoplasia and the probability that dogs with a negative cytology truly do not have a neoplasm, respectively. In this context, the aim of this study was to

determine the diagnostic accuracy of cytology in the diagnosis of canine splenic neoplasms utilizing the corresponding histopathology as the diagnostic reference standard [24,31,33,35,36], following STARD guidelines [34]. Additionally, sensitivity of cytology in the diagnosis of specific tumor types and in the identification of nodular versus diffuse neoplasms was evaluated.

## Materials and methods

### Criteria of selection of cases

In this retrospective study, the electronic cytological database of the Diagnostic Pathology Service of the Department of Veterinary Medicine (DIMEVET) of the University of Milan was searched for splenic samples collected between January 1st 1998 to January 31st 2018. The database was searched for specific key words in the following combinations: 1) "dog" and "spleen", 2) "dog" and "splenic". A consecutive series of canine splenic cytologies was obtained. Cytological samples came from external referring private practices or from the Veterinary Teaching Hospital (VTH) of the DIMEVET, or were prepared from fresh surgical biopsies and necropsies. Samples were submitted or collected to evaluate splenomegaly or nodular lesions.

The histopathology database was then searched for the histopathology corresponding to the same lesion examined by cytology. Histopathological samples were obtained from splenic biopsies (nodular lesions) or whole spleens (from splenectomies or necropsies) submitted to the Diagnostic Pathology Service of the DIMEVET, or were represented by slides submitted by external pathologists as second opinion cases. A time interval >45 days between cytological and histological sampling was an exclusion criterion. Histopathological samples collected via needle core biopsies were excluded from the study as they may bear reduced diagnostic reliability compared to incisional and excisional histological samples [8,37]. Cases were included in the study only when cytological and histological slides of the same lesion were available for review.

Additional information collected from the archives for cases included in the study were: sex, age, breed, cytological sampling technique (e.g. fine needle aspiration–FNA, touch imprint, scraping) and gross appearance of the lesion (i.e. diffuse versus nodular lesion).

To improve data completeness, transparency, and reproducibility, the study was conducted following the STARD guidelines [34] to the best of authors' ability. All canine splenic samples included in the current retrospective work, regardless of the sampling technique applied, were submitted to the Diagnostic Pathology Service of the DIMEVET for diagnostic purposes of spontaneous developing diseases. No animals were sampled or euthanized for research use. The use of animal tissue in the current study was approved by the Ethics Committee in charge for animal welfare of the University of Milan (Organismo Preposto al Benessere degli Animali, OPBA) with protocol number OPBA_86_2019. Sensitive information regarding owners and animals were stored, managed and preserved according to European and Italian laws.

### Sample processing

Cytological samples were air dried and stained with May-Grünwald-Giemsa (Merck KGaA, Frankfurt, Germany). Tissue samples for histopathology were fixed in 10% neutral buffered formalin, processed routinely, and embedded in paraffin wax. Sections of 5 μm were stained with Hematoxylin and Eosin. Second opinion cases were provided as Hematoxylin and Eosin stained slides by the referring pathologists.

## Case review

All cytological and histopathological samples were independently reviewed in a blinded fashion by three cytologists (M.C.—ECVP, G.G.—ECVCP, M.G.—resident) and by three anatomical pathologists (A.F.–ECVP, P.R.—ECVP, M.T.—resident), respectively. Both cytologists and anatomical pathologists were blinded to signalment information related to each case.

For each cytological case, one slide for each sampling technique was reviewed. First, slides were examined at low-power magnification (i.e. 10x objective lens) to assess the adequacy of the specimen. Poorly cellular samples were those characterized by marked hemodilution in the absence of both stromal elements and a mixed population of leukocytes [30]. Poorly cellular samples, poorly smeared samples (i.e. too thick or where most cells were ruptured), and samples where stain quality impaired adequate definition of the cell type (e.g. formalin-contaminated smears), were considered inconclusive [30]. Inconclusive cases were excluded from the statistical analysis as previously reported [8,24,32,33,35–37].

Cytological diagnoses were expressed according to those reported in the literature [4,30,38,39]. To facilitate comparison of the agreement between cytological and histological results, each cytological sample was further classified as neoplastic or non-neoplastic according to the main pathologic process. Non-neoplastic samples were those characterized by degenerative, reactive (including extramedullary hematopoiesis) [23,30], and inflammatory changes, as well as normal specimens consisting of stromal elements with mixed leukocyte population [30,38]. Reviewing cytologists were not allowed to use diagnostic modifiers such as "probably", "most likely", "suggestive of", as previously reported [37], nor to provide equivocal diagnoses (i.e. reporting more than one differential diagnosis). When a univocal diagnosis was not reached, cytologists reviewed the case collaboratively to find an agreement. Only the definitive diagnosis was included in the consecutive statistical analysis.

Neoplastic cytological samples were further subdivided by tumor type into the following subcategories: benign soft tissue tumor (BSTT) including angioma, angiosarcoma (HES), soft tissue sarcoma (STS) excluding angiosarcoma, lymphoma (LYM), mast cell tumor (MCT), histiocytic tumors including hemophagocytic sarcoma (HS), other round cell tumors including plasma cell tumor, myeloid leukemia and undifferentiated round cell tumor (ORCT), carcinoma (CARC), malignant neoplasm not otherwise specified (MNNOS).

Cases were included only when the three anatomical pathologists were in agreement because histopathology served as the diagnostic reference standard to evaluate cytological diagnostic accuracy. Neoplasms were classified applying the World Health Organization's histologic classification of tumors in domestic animals [40–48]. To further standardize histopathological diagnoses, anatomic pathologists were invited to classify some specific pathological entities (i.e. lymphomas, histiocytic proliferative disorders, nodular lesions previously classified as "fibrohistiocytic nodules") according to criteria reported in recent literature [49–52]. For nodular lesions, histological samples were considered conclusive and therefore only included in the statistical analysis if at least one slide containing at least one margin between the nodule and the adjacent splenic parenchyma was available for review [2,3,5].

Histological samples were then classified as neoplastic and non-neoplastic. Neoplastic cases were further subdivided utilizing the same subcategories applied to cytological samples.

## Data analysis

For all cases, the cytological diagnosis was compared with its paired histopathological diagnosis. Since histological samples maintain tissue architecture and are not biased by cellularity [8,32,37,53], histopathology was set as the reference standard as previously reported [24,31,33,35,36].

To determine diagnostic accuracy indexes, cytological specimens were classified according to four correlation categories (true positive, true negative, false positive, false negative). Specifically, the True Positive (TP) category included all cytological samples diagnosed as neoplastic with a corresponding neoplastic histopathology. The True Negative (TN) category comprised all cytological samples diagnosed as non-neoplastic with a corresponding non-neoplastic histopathology. The False Positive (FP) category included all cytological samples diagnosed as neoplastic with a corresponding non-neoplastic histopathology. The False Negative (FN) category comprised all non-neoplastic cytological diagnoses with a corresponding neoplastic histopathology.

To evaluate the sensitivity of cytology in the diagnosis of specific tumor types, only those cases histologically confirmed as neoplastic were taken into account. The subcategories assigned to each cytological and corresponding histological sample were then compared. When cytological and histological diagnoses matched for both neoplastic categorization and tumor type subcategorization, the case was defined as "true positive with complete agreement". When cytological and histological diagnoses matched for the neoplastic categorization but did not match for the tumor type subcategorization, the case was defined as "true positive with partial agreement". When a histopathological diagnosis categorized as neoplastic corresponded to a cytological diagnosis categorized as non-neoplastic, the case was considered in disagreement and defined as "false negative case".

## Statistical methods

Cytological-histological correlation categories (TP, FP, TN, FN) were included in a 2x2 table and used to calculate point estimates of overall accuracy, sensitivity, specificity, positive and negative predictive values [36,54]. Overall accuracy was defined as the ability of cytology to correctly identify neoplastic and non-neoplastic lesions, and was calculated as the sum of cases in which cytology and histology agreed in diagnosing a lesion as neoplastic (i.e. TP) or non-neoplastic (i.e. TN), divided by the total number of cases included in the study [32,55]. Given that pre-determined acceptability criteria for diagnostic performance of splenic cytology to distinguish between neoplastic and non-neoplastic lesions have not been previously established, overall accuracy, sensitivity, specificity, positive and negative predictive values were considered low if <70%, moderate if ≥70% and <80%, high if ≥80% and <90%, and very high if ≥90% [54]. To increase data comparability with other studies, positive and negative likelihood ratios were calculated [36]. Ninety-five percent (95%) confidence interval was calculated for each of the above mentioned indices of diagnostic test accuracy using a web-based application (MEDCALC—https://www.medcalc.org/calc/diagnostic_test.php).

The level of agreement between cytology and histopathology in the diagnosis of splenic neoplastic conditions was further investigated calculating the Cohen's kappa coefficient (κ), which was then corrected by the standard error. The value of k can be indicative of no agreement (if k <0), slight agreement (k = 0–0.20), fair agreement (k = 0.21–0.40), moderate agreement (k = 0.41–0.60), substantial agreement (k = 0.61–0.80), almost perfect agreement (k = 0.81–0.99), or perfect agreement (k = 1) [36,54]. Cohen's kappa and standard error were calculated utilizing GraphPad QuickCalcs Web site (GraphPad Inc.—https://www.graphpad.com/quickcalcs/kappa2/).

Sensitivity of cytology in differentiating splenic tumor types was defined as the ability of cytology to correctly identify as neoplastic a sample belonging to a specific neoplastic subcategory. Therefore, sensitivity for each tumor type was calculated as the sum of cases in complete and partial agreement (i.e. true positive cases) divided by the total number of cases with that specific neoplasm [31].

Similarly, the sensitivity of cytology in the diagnosis of neoplastic lesions according to their distribution pattern (i.e. diffuse or nodular) was evaluated. For each distribution pattern, sensitivity was calculated as the sum of cases in complete and partial agreement (i.e. true positive) divided by the total number of cases with a specific distribution pattern. Sensitivity of cytology according to distribution pattern was calculated for neoplastic lesions in general (i.e. the general sensitivity value obtained in our study), as well as for those specific neoplastic subcategories including cases with either diffuse or nodular distribution pattern.

Chi-square analysis applied to pairwise comparison was performed to evaluate whether statistically significant differences existed in the sensitivity of cytology for different tumor types, as well as in the diagnosis of neoplastic lesions with diffuse or nodular distribution pattern [36,37]. Specifically, the sensitivity of cytology for each neoplastic subcategory was compared with the sensitivity for splenic neoplasms in general, the sensitivity for all other neoplastic subcategories, and the sensitivity for any other neoplastic subcategory. Similarly, the difference between sensitivity for nodular or diffuse lesions among neoplasm in general and for each neoplastic subcategory was statistically investigated. Chi-square analysis was performed only on sensitivity values different from 0% and 100%, using MEDCALC (https://www.medcalc. org/calc/comparison_of_proportions.php). A *p*-value <.05 was considered statistically significant.

## Results

### Animals and samples

From a total of 950 splenic cytological samples retrieved between 1998–2018, 92 cytological samples from 91 dogs were included in the study; one dog was sampled for two distinct splenic lesions. A total of 858 splenic cytological cases were excluded for the following reasons: lack of a corresponding histopathological sample (832 cases), unavailable cytological and/or histological samples (16 cases), and needle core biopsies (10 cases). Among the selected cytological cases, 14 were considered inconclusive, and therefore excluded from the consecutive statistical analysis. Detailed evaluation of diagnostic accuracy was performed on 78/92 reviewed cytological samples (retrieval rate: 84.78%) obtained from 77 dogs [37].

Sex was available for 76/77 dogs: 19 were spayed females, 6 neutered males, 21 intact females, and 30 intact males. Mean age was 9.05 years (age range 2 months-16 years; age was not available for 2 cases). Twenty-six (26) breeds other than mongrels were represented; in one case breed was not provided.

The time interval between cytological sampling and corresponding histopathology collection ranged from 0 to 44 days for all cases.

Of the 78 cases included in the study, 81 cytological slides were evaluated (3 cases were sampled with two different techniques, i.e. touch imprinting and scraping). Cytological samples consisted of 43/81 touch imprints (53.09%) collected from both surgical biopsies and necropsies, 28 FNAs (34.57%), of these 21 were ultrasound guided, 1 was CT-scan guided, 1 was obtained during surgery, while in 5 FNA biopsies no additional sampling information was available. In 6 cases scrapings were obtained from surgical and necropsy specimens (7.41%). In 4 cases (4.94%) the sampling technique was not specified.

Complete agreement among anatomical pathologists was reached for all the 78 corresponding histopathological samples. Histopathological specimens were distributed as follows: 51 surgical samples from partial or complete splenectomies (51/78 cases, 65.38%), 24 spleens from necropsies (24/78, 30.77%), and 3 cases submitted as a second opinion (3/78, 3.85%).

**Table 1. Prevalence, agreement levels and sensitivity of cytology in the diagnosis of each neoplastic subcategory.**

|  | Prevalence | TP cases with complete agreement | TP cases with partial agreement | FN cases | Sensitivity | Confidence Interval (95%) |
|---|---|---|---|---|---|---|
| **TOTAL** | 71.79% (56/78) | 42.86% (24/56) | 21.43% (12/56) | 35.71% (20/56) | 64.29% | 50.36%–76.64% |
| **HES** | 28.57% (16/56) | 68.75% (11/16) | 6.25% (1/16) | 25% (4/16) | 75% | 47.62%–92.73% |
| **LYM** | 28.57% (16/56) | 37.50% (2/16) | 12.50% (6/16) | 50% (8/16) | 50% | 24.65%–75.35% |
| **STS** | 12.50% (7/56) | 42.86% (3/7) | 28.57% (2/7) | 28.57% (2/7) | 71.43% | 29.04%–96.33% |
| **BSTT** | 8.93% (5/56) | 0% (0/5) | 0% (0/5) | 100% (5/5) | 0% | 0.00%–52.18% |
| **HS** | 7.14% (4/56) | 25% (1/4) | 50% (1/4) | 25% (1/4) | 75% | 19.41%–99.37% |
| **MCT** | 7.14% (4/56) | 100% (4/4) | 0% (0/4) | 0% (0/4) | 100% | 39.76%–100% |
| **CARC** | 5.36%(3/56) | 66.67% (2/3) | 33.33% (1/3) | 0% (0/3) | 100% | 29.24%–100% |
| **ORCT** | 1.79% (1/56) | 0% (0/1) | 100% (1/1) | 0% (0/1) | 100% | 2.50%–100% |

BSTT, benign soft tissue tumor including angioma; CARC, carcinoma; FN, false negative; HES, angiosarcoma; HS, histiocytic neoplasm (including hemophagocytic syndrome); LYM, lymphoma; MCT, mast cell tumor (MCT); ORCT, other round cell tumor; STS, soft tissue sarcoma other than angiosarcoma; TP, true positive.

## Cytological and histological diagnoses

All cytological and corresponding histopathological diagnoses (78 cases) are listed in S1 Table. The diagnoses for the cytological-histological pairs excluded due to inconclusive cytology are listed in S2 Table.

No diagnostic differences were found for samples collected using two different sampling techniques, and therefore they were considered as one case in the consecutive statistical analysis. Cytologically, 37/78 cases were diagnosed as neoplastic (47.44%) and 41/78 as non-neoplastic (52.56%). All cases diagnosed as neoplastic were classified as malignant, and indeed, no benign neoplasms were cytologically observed.

Histologically, 56/78 cases (71.79%) were neoplastic (S1 Table) and 22/78 cases (28.21%) were non-neoplastic. Malignant tumors were 51 (51/56 tumors, 91.07%) and 5 were benign. The prevalence of each tumor type is reported in Table 1. No malignant neoplasm not otherwise specified was included in the study.

Of the 78 splenic lesions, 60 were nodular (76.92%), and 17 were diffuse (21.79%), while no information regarding the distribution pattern was available for 1 case (1.28%). Of the 56 neoplastic lesions, 43/56 cases (76.79%) were nodular and 12 cases (21.43%) were diffuse. The case for which distribution pattern was not provided was a liposarcoma (1.79%). This case was excluded from the evaluation of cytology sensitivity according to neoplasm distribution pattern. The proportion of cases with nodular or diffuse pattern for each tumor type are reported in Table 2.

**Table 2. Prevalence, agreement levels and sensitivity of cytology in the diagnosis of each neoplastic subcategory on the basis of distribution pattern.**

|  | Nodular | | | | Diffuse | | | |
|---|---|---|---|---|---|---|---|---|
|  | # of cases | TP | FN | Sensitivity (95% CI) | # of cases | TP | FN | Sensitivity (95% CI) |
| **TOTAL** | 43/56 (76.79%) | 26/43 | 17/43 | 60.47% (44.41%–75.02%) | 12/56 (21.43%) | 9/12 | 3/12 | 75% (42.81%–94.51%) |
| **HES** | 14/16 (87.50%) | 10/14 | 4/14 | 71.43% (41.90%–91.61%) | 2/16 (12.50%) | 2/2 | 0/2 | 100% (15.81%–100%) |
| **LYM** | 10/16 (62.50%) | 5/10 | 5/10 | 50% (18.71%–81.29%) | 6/16 (37.50%) | 3/6 | 3/6 | 50% (11.81%–88.19%) |
| **MCT** | 1/4 (25%) | 1/1 | 0/1 | 100% (2.50%–100%) | 3/4 (75%) | 3/3 | 0/3 | 100% (29.24%–100%) |

CI, confidence interval; FN, false negative; HES, angiosarcoma; LYM, lymphoma; MCT, mast cell tumor (MCT); TP, true positive.

**Table 3. Cytological-histological correlation categories.**

| Diagnosis | Histology: neoplastic | Histology: non-neoplastic | Total |
|---|---|---|---|
| **Cytology: neoplastic** | 36 (TP) | 1 (FP) | 37 |
| **Cytology: non-neoplastic** | 20 (FN) | 21 (TN) | 41 |
| **Total** | 56 | 22 | 78 |

FN, false negative; FP, false positive; TN, true negative; TP, true positive.

## Cyto-histological correlation

Following the tabulation of cytological and histological diagnoses (S1 Table), 36 cases (46.15%) were classified as TP, 21 (26.92%) were TN, 20 (25.64%) were FN, and 1 (1.28%) was a FP (Table 3).

The FP case had a cytological diagnosis of lymphoma that corresponded histologically to a purulent bacterial splenitis (in this case the full spleen was available for analysis and no tumor was found; however, severe marginal zone hyperplasia was present).

Neoplastic and non-neoplastic lesions were correctly identified in 57/78 cases (Table 3), therefore overall accuracy of cytology was 73.08% (Table 4). Sensitivity of cytology in the diagnosis of splenic neoplasms was 64.29%, specificity was 95.45%, positive predictive value was 97.30%, and negative predictive value was 51.22% (Table 4). Positive and negative likelihood ratios were 14.14 and 0.37, respectively (Table 4).

According to Cohen's test the level of agreement was considered as "moderate", with a κ value of 0.473 corresponding to a standard error of 0.086.

The distribution of TP and FN cases for each tumor type is reported in Table 1. The sensitivity of cytology in the diagnosis of each neoplastic subcategory was 100% for MCT, CARC and ORCT, 75% for HES, 75% for HS, 71.43% for STS, 50% for LYM, and 0% for BSTT included in the study. Further details regarding complete and partial agreement between cytological and histological diagnoses as well as confidence intervals of sensitivity value for each tumor type are listed in Table 1. Chi-square analysis of cytological sensitivity was applicable only to HES, HS, STS, and LYM. No statistically significant sensitivity differences were observed (*p*-value ranging from 0.1506 to 1.0).

**Table 4. Prevalence of neoplastic lesions, with point estimate and 95% confidence interval of diagnostic accuracy indexes, likelihood ratios and Cohen's k.**

| Diagnostic accuracy index | Value | Confidence Interval (95%) |
|---|---|---|
| **Prevalence** | 71.79% | 60.47%–81.41% |
| **Overall accuracy** | 73.08% | 61.84%–82.50% |
| **Sensitivity** | 64.29% | 50.36%–76.64% |
| **Specificity** | 95.45% | 77.16%–99.88% |
| **PPV** | 97.30% | 84.01%–99.60% |
| **NPV** | 51.22% | 42.21%–60.15% |
| **PLR** | 14.14 | 2.06–96.94 |
| **NLR** | 0.37 | 0.26–0.54 |
| **K value** | 0.473 | 0.304–0.643 |

NLR, negative likelihood ratio; NPV, negative predictive value; PLR, positive likelihood ratio; PPV, positive predictive value.

The proportion of TP and FN cases with nodular or diffuse pattern for each tumor type are reported in Table 2. Sensitivity of cytology in the diagnosis of neoplastic lesions in general according to their distribution pattern was 60.47% for nodular and 75% for diffuse neoplasms, with no statistically significant difference between the two values ($p = 0.3593$). For some tumor types the sensitivity of cytology on the basis of the distribution pattern was not calculated, given that only nodular (BSTT, STS, HS, CARC) or diffuse (ORCT) neoplastic lesions were represented in these categories. Sensitivity in the diagnosis of nodular and diffuse lymphomas was for both 50%, with no statistically significant difference between the two values ($p = 1.0$). Sensitivity for nodular angiosarcomas was 71.43% and 100% for diffuse angiosarcomas, while sensitivity for both nodular and diffuse mast cell tumors was 100%. Considering these results, Chi-square analysis of sensitivity on the basis of the distribution pattern was not performed for angiosarcomas and mast cell tumors.

## Discussion

In this study we report overall accuracy, sensitivity, specificity, positive and negative predictive values of cytology for the diagnosis of canine splenic neoplasms. Similar studies [8,23–25,31–33] have limited the evaluation of cytological diagnostic accuracy to overall agreement with histopathology, hampering comparison with our results. Our study has evidenced a moderate overall accuracy of cytology. Specifically, although this technique had a high specificity and positive predictive value for the diagnosis of splenic neoplasia, sensitivity and negative predictive value were lower, indicating that cytological diagnosis of splenic neoplasia is reliable, but a negative result cannot be used to exclude the possibility of splenic neoplasia.

According to overall accuracy and Cohen's k values, cytology is not a reliable alternative to histopathology in the definitive diagnosis of splenic tumors in most cases. When compared with previous studies, our overall accuracy value (73.08%) laid in between the higher range of 83.87–100% [23–25] and the lower 38–69.7% range [8,31–33] reported in other studies. To allow comparison, the overall accuracy (intended as the sum of complete and partial diagnostic agreements) was calculated from the raw data of previously published caseloads [8,23–25,31–33] when not made explicit in the corresponding manuscript.

Low sensitivity and negative predictive value of this study indicate that a cytology negative for neoplasia should prompt further investigations to confirm a dog to be truly free from neoplastic disease. This contrasts with our initial hypothesis that cytology may represent a useful tool to avoid unnecessary splenectomy. Instead, high specificity and positive predictive value identify cytology as a good and reliable tool to rule in the diagnosis of splenic neoplasia with a high degree of confidence. In practical terms, a cytology positive for neoplasia may lead to a faster surgical treatment, avoiding lag times and higher costs associated with application of diagnostic imaging techniques such as contrast-enhanced ultrasound and computed tomography (CT) [1,56]. Our results are in line with studies evaluating diagnostic accuracy of cytology applied to various organs in dogs [31,32,35,36,54], with sensitivity and negative predictive value generally lower than specificity and positive predictive value, respectively.

Regarding the reliability of cytology in the diagnosis of specific tumor types, the lack of statistically significant differences between subcategories may be related to an imbalance in the number of cases for each tumor type. Also, our results may be influenced by the tumor cell type evaluated, since exfoliation rate varies substantially between round cell, epithelial and mesenchymal tumors [30,31,35,53]. Specifically, mesenchymal tumors have the lowest tendency to exfoliate [30,31,35,53] explaining the low sensitivity in diagnosing benign mesenchymal tumors. Moreover, identification of vascular tumors (i.e. angiomas and HES) among false

negative cases is not surprising since the architecture of these tumors often leads to significant peripheral blood contamination in aspirates [4,30,32,57]. The low sensitivity of cytology in the diagnosis of splenic lymphomas relates to the specific distribution of tumor types in the spleen, where indolent nodular lymphomas (i.e. mantle cell lymphoma and marginal zone lymphoma) are frequent as was in this caseload. These are nodular lymphomas composed of small to medium sized cells with minimal atypia and a low mitotic rate [49,58]. Thus, mantle cell lymphoma and marginal zone lymphoma can be easily misinterpreted as reactive lymphoid hyperplasia on cytology, and histopathology is often necessary for a definitive diagnosis that relies on the evaluation of tumor architecture [49].

Although not statistically significant, our results paralleled those of previous reports identifying higher cytological accuracy in the diagnosis of diffuse compared to focal lesions [23,32,57].

One false positive diagnosis of neoplasia (i.e. lymphoma) was included in this study. Splenic marginal zone hyperplasia is a common finding in dogs [49,59], and cytological sampling from these areas may result in a monomorphic specimen mimicking marginal zone lymphoma. This is a risk that pathologists have to bear in mind; thus the diagnosis of nodular low-grade lymphoma should be supported by histological evaluation of architectural changes, especially in dogs. A recent report [60] has demonstrated a high overall concordance between histopathology, immunohistochemistry and PCR for antigen receptor rearrangement (PARR) in the diagnosis of marginal zone lymphoma, mantle cell lymphoma and lymphoid or complex nodular hyperplasia. Therefore, further development of combined methods also applicable to cytological specimens may provide a less invasive and more valuable diagnostic approach to the diagnosis of splenic nodular lymphoid lesions.

Although histopathology is generally considered the diagnostic reference standard [24,31,33,35,36], several limitations should also be considered for this technique in the diagnosis of splenic tumors. Specifically, the diagnosis of splenic hematomas and hemangiosarcomas is considered difficult, especially if spleens are not submitted entirely and if adequate samples from the margin of the lesion are not collected [2,3,5]. Noteworthy, splenic hematomas and HES may not be grossly distinguishable [2,3,5,23,61], and the first may represent a component of the latter [61].

The current study is characterized by several limitations, mainly due to its retrospective nature. One major limit was the inclusion of specimens obtained by different sampling techniques, with a high number of impression smears collected from both surgical biopsies and necropsies. Additionally, the inclusion of splenic cytological samples from necropsies and the university setting of this work may have further biased the study toward cases with a more aggressive behavior and with features of malignancy easier to diagnose. This may not reflect daily clinical practice in which FNA is the most common sampling technique to pre-operatively assess splenic lesions. Also, as previously observed [31,53], different sampling methods may have resulted in an improvement of sensitivity of cytology in this study, especially for those neoplasms characterized by low exfoliation rate. On the other hand, this observation can be viewed also in positive terms. Indeed, in a practical setting the preliminary evaluation of surgical biopsies or entire spleens by cytology prior to fixation could be implemented to facilitate the diagnosis and to reduce turnaround time. Additionally, this approach can provide pathologists with material useful not only for a preliminary diagnosis, but also for immunocytochemistry and for PARR on fresh specimens.

Despite this caseload being larger than previously reported ones, the small number of cases evaluated may explain the relatively wide confidence intervals observed around point estimates of our diagnostic accuracy indexes. Results may have been further biased by the inclusion

criteria applied in the current study, leading to the exclusion of more than 90% cytological samples of canine spleen in our archives.

Unfortunately, full agreement with STARD guidelines could not be obtained in this study since the type of treatment administered between cytological and histological sampling, and the incidence of adverse events following splenic sampling, could not be retrieved from our electronic archives.

In conclusion, to the best of our knowledge, this is the first study conjunctively reporting overall accuracy, sensitivity, specificity, positive and negative predictive values of cytology in the diagnosis of canine splenic neoplasms compared to histopathology. Diagnostic accuracy indexes identified limitations of negative cytological results in excluding a dog to be truly free from neoplasia; however, high specificity and positive predictive value highlighted cytology as a valuable tool in the diagnostic approach to splenic neoplasms.

## Supporting information

**S1 Table. Cytological and corresponding histological diagnosis for each case included in the statistical analysis, with correlation category and lesion distribution pattern.** Neoplastic subcategory and level of agreement are provided when applicable. Dogs from which more than one sample was obtained are marked with (\*). The age of dogs is reported in years, if not otherwise specified. Abbreviations: B, biopsy; BSTT, benign soft tissue tumor including angioma; CA, complete agreement; CARC, carcinoma; DA, disagreement; EMH, extramedullary hematopoiesis; F, female; FN, false negative; FNA, fine needle aspirate; FP, false positive; HES, angiosarcoma; HS, histiocytic neoplasm (including hemophagocytic syndrome); LYM, lymphoma; M, male; MCT, mast cell tumor (MCT); mm, months; MNNOS, malignant neoplasm not otherwise specified; N, necropsy; NF, neutered female; NM, neutered male; NPL, neoplastic; NON-NPL, non-neoplastic; NOS, not otherwise specified; n/a, not applicable; n/d, not determined; ORCT, other round cell tumor; PA, partial agreement; RLH, reactive lymphoid hyperplasia; SC, scraping; STS, soft tissue sarcoma other than angiosarcoma; TN, true negative; TP, true positive; 2OP, second opion case.
(XLSX)

**S2 Table. Cytological and corresponding histological diagnosis for each cytologically inconclusive case, with lesion distribution pattern.** Neoplastic subcategory is provided when applicable. Abbreviations: B, biopsy; BSTT, benign soft tissue tumor including angioma; CARC, carcinoma; EMH, extramedullary hematopoiesis; F, female; FNA, fine needle aspirate; HES, angiosarcoma; LYM, lymphoma; M, male; N, necropsy; NF, neutered female; NM, neutered male; NPL, neoplastic; NON-NPL, non-neoplastic; n/a, not applicable; STS, soft tissue sarcoma other than angiosarcoma; 2OP, second opinion case.
(XLSX)

## Acknowledgments

We thank Professor Valerio Bronzo for theoretical assistance with the setting of Chi-square analysis.

## Author Contributions

**Conceptualization:** Annalisa Forlani, Mario Caniatti, Paola Roccabianca.

**Data curation:** Marco Tecilla, Matteo Gambini, Annalisa Forlani.

**Formal analysis:** Marco Tecilla, Matteo Gambini.

**Investigation:** Marco Tecilla, Matteo Gambini, Annalisa Forlani, Mario Caniatti, Gabriele Ghisleni, Paola Roccabianca.

**Methodology:** Matteo Gambini, Annalisa Forlani, Paola Roccabianca.

**Project administration:** Paola Roccabianca.

**Resources:** Annalisa Forlani, Mario Caniatti, Paola Roccabianca.

**Supervision:** Mario Caniatti, Paola Roccabianca.

**Validation:** Marco Tecilla, Matteo Gambini.

**Visualization:** Marco Tecilla, Matteo Gambini, Annalisa Forlani.

**Writing – original draft:** Marco Tecilla, Annalisa Forlani, Paola Roccabianca.

**Writing – review & editing:** Marco Tecilla, Matteo Gambini, Annalisa Forlani, Mario Caniatti, Gabriele Ghisleni, Paola Roccabianca.

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
