## [Decision Letter · Decision Letter 0]

7 Aug 2019

PONE-D-19-20551

Evaluation of cytological diagnostic accuracy for canine splenic neoplasms: an investigation in 78 cases on adherence to STARD guidelines.

PLOS ONE

Dear Dr. Gambini,

Thank you for submitting your manuscript to PLOS ONE. After careful consideration, we feel that it has merit but does not fully meet PLOS ONE’s publication criteria as it currently stands. Therefore, we invite you to submit a revised version of the manuscript that addresses the points raised during the review process.

Please address all Reviewer comments.

We would appreciate receiving your revised manuscript by Sep 21 2019 11:59PM. To enhance the reproducibility of your results, we recommend that if applicable you deposit your laboratory protocols in protocols.io, where a protocol can be assigned its own identifier (DOI) such that it can be cited independently in the future. For instructions see: http://journals.plos.org/plosone/s/submission-guidelines#loc-laboratory-protocols

We look forward to receiving your revised manuscript.

Kind regards,

Douglas H. Thamm, V.M.D.

Academic Editor

PLOS ONE

Journal Requirements:

The authors received no specific funding for this work.

We note that one or more of the authors are employed by a commercial company: IDEXX Laboratories, Wetherby, West Yorkshire, UK

Reviewers' comments:

Reviewer's Responses to Questions

**Comments to the Author**

1. Is the manuscript technically sound, and do the data support the conclusions?

Reviewer #1: Partly

Reviewer #2: Yes

2. Has the statistical analysis been performed appropriately and rigorously? 

Reviewer #1: Yes

Reviewer #2: Yes

3. Have the authors made all data underlying the findings in their manuscript fully available?

Reviewer #1: Yes

Reviewer #2: Yes

4. Is the manuscript presented in an intelligible fashion and written in standard English?

Reviewer #1: Yes

Reviewer #2: Yes

5. Review Comments to the Author

Reviewer #1: The paper is well-written, but the discussion should be shortened and rewritten to support the conclusions of the study rather than to reiterate results.

Minor suggestions:

Define STARD and DIMEVET in the abstract.

Add a space on line 39 of the abstract between of and 95.45%.

Line 48: Delete "fully".

Line 62: Delete "all".

Lines 69-70 Change to "...consisting mostly of controllable hemorrhage".

Line 90 Change to "...test with a high sensitivity..."

Line 141 Change to "Tissue samples for histopathology were fixed..."

Line 153 Change to "...(i.e., 10x objective lens) to..."

Line 171 Change "revised" to "reviewed"

Line 178 Change to "...other round cell tumors..."

Lines 229-230 Change to '...predetermined acceptability..."

Line 372 Should "p" be "p-value"?

Line 385 Change "Considered" to "Considering"

Line 417 Define "TC"

Line 488 Change "numerosity" to "number of cases evaluated"

Major question to consider:

Why were partial agreements included as true positives for statistical purposes when determining sensitivity of cytology in differentiating splenic tumor type subcategories?

Reviewer #2: The goals of this manuscript are to 1) systematically determine the diagnostic sensitivity and specificity of splenic aspiration in the diagnosis of neoplasia and 2) to ensure, to the best of the authors’ ability, that the study is carried out according to standardized guidelines for diagnostic accuracy studies (STARD). The authors have done a nice job of outlining existing weaknesses that exist in previous studies of canine splenic neoplasms and the role of cytology in making diagnoses. Original data are provided and methodology is provided in sufficient detail to understand how the study was completed.

Overall, this material provides some valuable data not specifically explored in previous manuscripts, and I think there are some interesting findings, such as differences in cytological diagnoses that are made from nodular or diffuse lesions. Conclusions are supported by provided data, and there are clear weaknesses in previous studies that are addressed by application of STARD criteria in this data set.

I have suggestions for some concepts and word choices that could benefit from additional clarification and/or expansion, as outlined below:

Concepts to clarify:

Line 59: I think this section could be worded a bit more strongly in order to emphasize the initial thought process behind your study. I’m not sure that a 52% survival post-splenectomy is “high.” I might say “fair” or simply list the overall survival rate, then go on to discuss complications related to splenectomy.

Paragraph, lines 76-89: This section could benefit from some reorganization for clarity. A few more details on the purpose of STARD guidelines would help readers understand what is unique about your study. For example, I might state the purpose of STARD and how guidelines came into existence (what are common weaknesses in studies of this kind that STARD addresses and improves?). Next, I would discuss where existing veterinary studies fall short. Finally, I would state that you are following recommended STARD criteria to perform cross-tabulation of test results against those of the reference standard to generate sensitivity and specificity data, absent in other studies.

Line 89: I’m not sure that I understand what a “deferred” cytological diagnosis is. Is that when cytology is equivocal until a histological diagnosis is available? If so, I think equivocal or provisional cover the spectrum of what is intended. If you mean “non-diagnostic,” say that instead.

Line 108: When you say “continuous series,” do you mean a series of diagnostic samples collected over 20 years, or a representative collection of splenic aspirate samples that include all pathologic states (neoplasia, hyperplasia, atypical hyperplasia, EMH, etc.)?

Line 128: Again, some expansion of exactly how your study is an improvement over others based on STARD guidelines would strengthen this section.

Paragraph, lines 161-173, Cytological diagnoses: Can you comment on the real world realism of removing diagnostic modifiers for these kinds of cases? I use diagnostic modifiers to improve the clarity of cytological reports, and in some cases, you might actually lose diagnostic information by being so stringent with reporting guidelines. I understand why the guidelines were set in this way for the study. However, readers may wonder what would happen if you had considered cases that originally included “most likely,” “suggestive of,” or “probable” HES at the time of diagnosis. Did you consider this, and based on the data you were able to evaluate, do you think this would this likely increase diagnostic sensitivity in “the real world” or would not actually have much of an effect?

Line 181: How often were all four anatomical pathologists in agreement for the reference standard, and how many cases were discarded? Did the discarded cases have interesting features pathologists should be aware of?

Line 220: You list a false positive in the conclusions, so it would be helpful to define “false positive” in this section alongside your other categories.

Line 492: You discuss that you were not able to completely follow STARD guidelines, but line 128 definitively states “The study was conducted following STARD guidelines.” Your disclaimer should probably come earlier in the manuscript so that there is not an apparent conflict in language.

Suggested language: Some language changes may improve readability. Here are some suggestions that may be helpful.

• Title: The way this is worded, it makes it sound like you are trying to determine how well your cases adhere to STARD guidelines, rather than determining diagnostic accuracy of cytology utilizing the STARD guidelines. Consider “Evaluation of cytological diagnostic accuracy for 78 canine splenic neoplasms using STARD guidelines.”

• Line 25: Cytology represents a useful diagnostic tool in the preliminary clinical 26 approach to canine splenic lesions, and may prevent unnecessary splenectomy.

• Line 42: When positive for neoplasia, cytology represents a useful diagnostic tool to rule in splenic neoplasia, prompting surgery independently from other diagnostic tests.

• Line 52: Use “fewer” instead of “less,” since the total quantity of splenic lesions diagnosed as HES or other is actually known.

• Line 69: Again use “fewer” instead of “less” since the number of total complications is known.

• Line 78: Word choice for clarity – consider “comprehensively assessed” instead of “addressed conjunctively”

• Line 166: Are reactive and inflammatory separate categories? If so, we need a comma after “reactive (including extramedullary hematopoiesis”)

• Line 169: remove the word “with.”

• Line 190: Move “only” to before “included” to improve flow.

• Line 237: Use “were” instead of “was”

• Line 230: Use “pre-determined” instead of “pre-determine”

• Line 312: Use “and indeed,”

• Line 385: Use “Considering”

• Line 426: Use “imbalance” instead of “unbalance”

6. PLOS authors have the option to publish the peer review history of their article (what does this mean?). If published, this will include your full peer review and any attached files.

Reviewer #1: No

Reviewer #2: Yes: Lisa J. Schlein

---

## [Author Response · Author response to Decision Letter 0]

17 Sep 2019

REBUTTAL LETTER – Manuscript PONE-D-19-20551

RESPONSES TO REVIEWER 1 

1) “The paper is well-written, but the discussion should be shortened and rewritten to support the conclusions of the study rather than to reiterate results.” 

As suggested, the discussion has been shortened and reworded to avoid as possible reiteration of results. Major changes were avoided to try to balance Reviewer 1 and Reviewer 2 observations regarding the discussion. 

 The following changes have been implemented:

- Former lines 399-408 (current lines 416-423) now run as follows: “According to overall accuracy and Cohen’s k values, cytology is not a reliable alternative to histopathology in the definitive diagnosis of splenic tumors in most cases. When compared with previous studies, our overall accuracy value (73.08%) laid in between the higher range of 83.87-100% [23–25] and the lower 38-69.7% range [8,31–33] reported in other studies. To allow comparison, the overall accuracy (intended as the sum of complete and partial diagnostic agreements) was calculated from the raw data of previously published caseloads [8,23–25,31–33] when not made explicit in the corresponding manuscript.”

- Former lines 421-427 (current lines 437-439) now run as follows: “Regarding the reliability of cytology in the diagnosis of specific tumor types, the lack of statistically significant differences between subcategories may be related to an imbalance in the number of cases for each tumor type.”

- Former lines 443-447 (current lines 455-457) now run as follows: “Although not statistically significant, our results paralleled those of previous reports identifying higher cytological accuracy in the diagnosis of diffuse compared to focal lesions [23,32,57].“

- Former lines 448-450 (current lines 458-459) now run as follows: “One false positive diagnosis of neoplasia (i.e. lymphoma) was included in this study.”

2) “Define STARD and DIMEVET in the abstract.”

Changed as suggested (current lines 31 and 33-34). Additionally, to maintain consistency with the Material and Methods section (current line 118), “Università degli Studi di Milano” was changed in “University of Milan” (current line 34).

3) “Add a space on line 39 of the abstract between of and 95.45%.”

Changed as suggested (current line 40).

4) “Line 48: Delete "fully".”

Changed as suggested (current lines 50-51).

5) “Line 62: Delete "all".”

Changed as suggested (current line 68).

6) “Lines 69-70 Change to "...consisting mostly of controllable hemorrhage".”

Changed as suggested (current line 76).

7) “Line 90 Change to "...test with a high sensitivity..."”

Changed as suggested (current line 104).

8) “Line 141 Change to "Tissue samples for histopathology were fixed..."”

Changed as suggested (current line 155).

9) “Line 153 Change to "...(i.e., 10x objective lens) to..."”

Changed as suggested (current lines 167-168).

10) “Line 171 Change "revised" to "reviewed"”

Changed as suggested (current lines 185).

11) “Line 178 Change to "...other round cell tumors..."”

Changed as suggested (current lines 192).

12) “Lines 229-230 Change to '...predetermined acceptability..."”

Changed as suggested (current lines 245-246).

13) “Line 372 Should "p" be "p-value"?”

“p” has been changed to “p-value” (current line 287). Additionally, the letter “p” has been changed to italic throughout the manuscript (current lines 389, 395 and 400) in agreement with papers previously published by PLOS ONE. 

14) “Line 385 Change "Considered" to "Considering"”

Changed as suggested (current line 402).

15) “Line 417 Define "TC"”

“TC” was a typo. The sentence (current lines 431-433) now runs as follows: “…avoiding lag times and higher costs associated with application of diagnostic imaging techniques such as contrast-enhanced ultrasound and computed tomography (CT) [1,56].”

16) “Line 488 Change "numerosity" to "number of cases evaluated"”

Changed as suggested (current lines 495-496).

17) “Why were partial agreements included as true positives for statistical purposes when determining sensitivity of cytology in differentiating splenic tumor type subcategories?”

This approach is consistent with criteria applied in previous studies (see: Cohen M, Bohling MW, Wright JC, Welles EA, Spano JS. Evaluation of sensitivity and specificity of cytologic examination: 269 cases (1999-2000). J Am Vet Med Assoc. 2003;222: 964–7). Indeed, the major aim of the study was to investigate the diagnostic accuracy of cytology in the diagnosis of splenic neoplasm. Based on this aim, the main threshold set was the ability of cytology to correctly identify the neoplastic nature of a lesion, against degenerative and inflammatory conditions. Since the cytological diagnosis of a neoplasm is considered as sufficient to elect for splenectomy, cases with “partial agreement” (defined as agreement on the main ongoing pathological process, intended as neoplastic versus non-neoplastic) were included in the statistical analysis as true positives. 

 To improve clarity, the following changes have been made in the Material and methods section:

- Former lines 212-220 (current lines 226-236) now run as follows: “To evaluate the sensitivity of cytology in the diagnosis of specific tumor types, only those cases histologically confirmed as neoplastic were taken into account. The subcategories assigned to each cytological and corresponding histological sample were then compared. When cytological and histological diagnoses matched for both neoplastic categorization and tumor type subcategorization, the case was defined as “true positive with complete agreement”. When cytological and histological diagnoses matched for the neoplastic categorization but did not match for the tumor type subcategorization, the case was defined as “true positive with partial agreement”. When a histopathological diagnosis categorized as neoplastic corresponded to a cytological diagnosis categorized as non-neoplastic, the case was considered in disagreement and defined as “false negative case”.”

- Former lines 248-251 (current lines 264-268) now run as follows: “Sensitivity of cytology in differentiating splenic tumor types was defined as the ability of cytology to correctly identify as neoplastic a sample belonging to a specific neoplastic subcategory. Therefore, sensitivity for each tumor type was calculated as the sum of cases in complete and partial agreement (i.e. true positive cases) divided by the total number of cases with that specific neoplasm [31].”

Note that percentage of cases in complete agreement (grouped by each tumor type in the third column of Table 1 (labeled as “TP cases with complete agreement“) completely overlaps with the value of sensitivity of cytology in recognizing a neoplasm as belonging to a specific neoplastic subcategory. 

RESPONSES TO REVIEWER 2

1) “Line 59: I think this section could be worded a bit more strongly in order to emphasize the initial thought process behind your study. I’m not sure that a 52% survival post-splenectomy is “high.” I might say “fair” or simply list the overall survival rate, then go on to discuss complications related to splenectomy.”

According to Reviewer 2’s request, former lines 57-62 have been rearranged and now run as follows (current lines 59-64): “While the two-month post-splenectomy survival rate is lower in dogs with HES (32%) compared to dogs with hematomas (83%), the general survival rate after splenectomy is 52%, regardless of underlying splenic pathology [3]. While this can be interpreted as fair survival data, a proportion of dogs (7.6%) will develop complications secondary to splenectomy because of thrombotic or coagulopathic syndromes [9].”

As suggested, complications following splenectomy have been included as follows (current lines 64-68): “Additional adverse effects following splenectomy in dogs have included peri- and post-operative ventricular arrhythmias [9–11], reduced blood filtration and renewal [12,13], impairment of humoral immune response [14], reduced immune-surveillance against bacteria and parasites [15–19], and higher incidence of gastric dilatation-volvulus [6,20–22].” 

Reference list has been updated accordingly. 

2) “Paragraph, lines 76-89: This section could benefit from some reorganization for clarity. A few more details on the purpose of STARD guidelines would help readers understand what is unique about your study. For example, I might state the purpose of STARD and how guidelines came into existence (what are common weaknesses in studies of this kind that STARD addresses and improves?). Next, I would discuss where existing veterinary studies fall short. Finally, I would state that you are following recommended STARD criteria to perform cross-tabulation of test results against those of the reference standard to generate sensitivity and specificity data, absent in other studies.”

According to Reviewer 2’s request, current lines 87-90 now run as follows: “STARD guidelines have been created to avoid incomplete reporting in diagnostic accuracy studies and to improve the general quality of the latter, reducing problems related to study identification, critical appraisal, and replication [34].“ 

According to Reviewer 2’s suggestion, the section was further expanded and current lines 98-102 now run as follows: “Application of STARD guidelines in the current study allowed cross-tabulation of cytological results (i.e. the index test) against those of histopathology (i.e. the reference standard) to generate sensitivity and specificity data [34], not retrieved from other former studies and useful to facilitate data comparability by potential future studies on the same topic.”

3) “Line 89: I’m not sure that I understand what a “deferred” cytological diagnosis is. Is that when cytology is equivocal until a histological diagnosis is available? If so, I think equivocal or provisional cover the spectrum of what is intended. If you mean “non-diagnostic,” say that instead.”

The terms came from previous reports. “Deferred” means that diagnoses were postponed awaiting histopathology. This does not mean neither equivocal nor provisional, but pending histology. This is somewhat an unusual way of dealing with cases, but this is how they were reported in previous reports. The closes term would be “provisional”. Thus “deferred” was erased and according to Reviewer 2’s suggestion, current lines 94-98 now run as follows: “In most reports, sensitivity, specificity, positive and negative predictive values of splenic cytology are not calculated and cannot be properly estimated [8,23–25,31–33] because caseloads simultaneously include multiple species [24,25,31–33], multiple tissues and organs [31–33], or “equivocal” or “provisional” cytological and histological diagnoses [23,24,33].”

4) “Line 108: When you say “continuous series,” do you mean a series of diagnostic samples collected over 20 years, or a representative collection of splenic aspirate samples that include all pathologic states (neoplasia, hyperplasia, atypical hyperplasia, EMH, etc.)?”

Using the term “continuous series” we meant a series of diagnostic samples collected over 20 years. Seen this observation and and according with STARD guidelines (including the specific explanation and elaboration document that can be found at: https://www.ncbi.nlm.nih.gov/pubmed/28137831), we substituted “continuous” with “consecutive” to improve clarity. Current lines 120-121 now run as follows: “A consecutive series of canine splenic cytologies was obtained.”

5) “Line 128: Again, some expansion of exactly how your study is an improvement over others based on STARD guidelines would strengthen this section.”

According to Reviewer 2’s suggestion, current lines 141-142 now run as follows: “To improve data completeness, transparency, and reproducibility, the study was conducted following the STARD guidelines [34] to the best of authors’ ability.” To avoid excessive expansion of the Materials and methods section, the explanation of usefulness and improvement provide by STARD guidelines were included in the Introduction as Reviewer 2 suggested (see answer to comment #2 by Reviewer 2). 

6) “Paragraph, lines 161-173, Cytological diagnoses: Can you comment on the real world realism of removing diagnostic modifiers for these kinds of cases? I use diagnostic modifiers to improve the clarity of cytological reports, and in some cases, you might actually lose diagnostic information by being so stringent with reporting guidelines. I understand why the guidelines were set in this way for the study. However, readers may wonder what would happen if you had considered cases that originally included “most likely,” “suggestive of,” or “probable” HES at the time of diagnosis. Did you consider this, and based on the data you were able to evaluate, do you think this would this likely increase diagnostic sensitivity in “the real world” or would not actually have much of an effect?”

This is a truly interesting point. Despite the setting of a scientific manuscript requires rigid categorization to perform clear cut statistics, we agree with Reviewer 2, and we usually include diagnostic modifiers in our routine cytological reports to express the degree of certainty in the diagnosis and/or the interpretation of the examined cytological samples. Removing such modifiers might, in some cases, implicate a loss of diagnostic information, intended as information that might have a considerable influence on clinical decision-making process following the cytological report. Therefore, diagnostic modifiers are useful and for sure they should not be avoided in real life scenarios. 

On the other hand, according to different reports, the use of modifiers and their meaning are not standardized, becoming subject to interpretation (see Sharkey LC, Dial SM, Matz ME. Maximizing the Diagnostic Value of Cytology in Small Animal Practice. Vet Clin North Am - Small Anim Pract. 2007;37: 351–372; and Christopher MM, Hotz CS. Cytologic diagnosis: expression of probability by clinical pathologists. Vet Clin Path. 2004;33(2): 84-95). In this specific context, the use of non-standardized parameters would introduce variability that could reduce or void the strength of the statistical analysis. Therefore, considered that the aim our work was to investigate the agreement of cytology (i.e. the index test) with histopathology (i.e. the reference test) disregarding the clinical history (another unreal scenario that we are asked for in papers and in board exams), and considered that an objective statistical analysis could not be performed if modifiers were included, we opted to remove modifiers during the review process of cytological samples. This choice is in line with other studies concerning diagnostic accuracy of cytology following STARD guidelines (see Berzina I, Sharkey LC, Matise I, Kramek B. Correlation between cytologic and histopathologic diagnoses of bone lesions in dogs: a study of the diagnostic accuracy of bone cytology. Vet Clin Pathol. 2008;37: 332–338.). 

Regarding the sentence stating that “readers may wonder what would happen if you had considered cases that originally included “most likely,” “suggestive of,” or “probable” HES at the time of diagnosis”, we fear that some kind of misunderstanding happened. Indeed, as stated in the Materials and methods section (current lines 116-121), samples included in the study belong to a consecutive series of archived cytological slides, mined using keywords only concerning the species and the organ from which cytological samples were obtained. This means that the original cytological diagnosis reported in our electronic archives was not a selection criterion for inclusion of cases in the study. Therefore, all the cytological samples conforming to the inclusion criteria (i.e. availability for review of both cytological slides and histopathological sample of the same lesion) were included in the study, disregarding the fact that the original diagnosis might be reported as “most likely”, “suggestive of”, or “probably” indicative of a lesion. Therefore, seen our selection criteria, the sensitivity should probably not change substantially. 

7) “Line 181: How often were all four anatomical pathologists in agreement for the reference standard, and how many cases were discarded? Did the discarded cases have interesting features pathologists should be aware of?”

Agreement among anatomical pathologists was reached for all the cases included in the study, as already stated in former lines 298-299 (current lines 315-316). 

Thanks to comment #7 by Reviewer 2, we found an error that has now been corrected throughout the manuscript. Specifically, former lines 149-150 reported that a fourth anatomical pathologist was involved in the study, but pathologists were actually three The following changes have been applied to correct the mistake:

- Current lines 161-164 (former lines 147-150): “All cytological and histopathological samples were independently reviewed in a blinded fashion by three cytologists (MC - ECVP, GG - ECVCP, MG - resident) and by three anatomical pathologists (AF – ECVP, PR - ECVP, MT - resident), respectively.”

- Current lines 195-197 (former lines 181-183): “Cases were included only when the three anatomical pathologists were in agreement because histopathology served as the diagnostic reference standard to evaluate cytological diagnostic accuracy.”

8) “Line 220: You list a false positive in the conclusions, so it would be helpful to define “false positive” in this section alongside your other categories.”

The definition of false positive cases is already available in the manuscript in the previous sentence lines (current lines 222-223). For better clarity, the text was modified (current lines 226-236) and now run as follows: “To evaluate the sensitivity of cytology in the diagnosis of specific tumor types, only those cases histologically confirmed as neoplastic were taken into account. The subcategories assigned to each cytological and corresponding histological sample were then compared. When cytological and histological diagnoses matched for both neoplastic categorization and tumor type subcategorization, the case was defined as “true positive with complete agreement”. When cytological and histological diagnoses matched for the neoplastic categorization but did not match for the tumor type subcategorization, the case was defined as “true positive with partial agreement”. When a histopathological diagnosis categorized as neoplastic corresponded to a cytological diagnosis categorized as non-neoplastic, the case was considered in disagreement and defined as “false negative case”.”

9) “Line 492: You discuss that you were not able to completely follow STARD guidelines, but line 128 definitively states “The study was conducted following STARD guidelines.” Your disclaimer should probably come earlier in the manuscript so that there is not an apparent conflict in language.”

According to Reviewer 2’s request, current lines 141-142 now run as follows: “To improve data completeness, transparency, and reproducibility, the study was conducted following the STARD guidelines [34] to the best of authors’ ability.”

10) “Title: The way this is worded, it makes it sound like you are trying to determine how well your cases adhere to STARD guidelines, rather than determining diagnostic accuracy of cytology utilizing the STARD guidelines. Consider “Evaluation of cytological diagnostic accuracy for 78 canine splenic neoplasms using STARD guidelines.””

The title (current lines 4-6) has been changed as suggested. 

11) “Line 25: Cytology represents a useful diagnostic tool in the preliminary clinical 26 approach to canine splenic lesions, and may prevent unnecessary splenectomy.”

Changed as suggested (current lines 25-27).

12) “Line 42: When positive for neoplasia, cytology represents a useful diagnostic tool to rule in splenic neoplasia, prompting surgery independently from other diagnostic tests.”

Changed as suggested (current lines 43-45).

13) “Line 52: Use “fewer” instead of “less,” since the total quantity of splenic lesions diagnosed as HES or other is actually known.”

Changed as suggested (current lines 54).

14) “Line 69: Again use “fewer” instead of “less” since the number of total complications is known.”

Changed as suggested (current lines 75).

15) “Line 78: Word choice for clarity – consider “comprehensively assessed” instead of “addressed conjunctively””

Changed as suggested (current lines 84).

16) “Line 166: Are reactive and inflammatory separate categories? If so, we need a comma after “reactive (including extramedullary hematopoiesis”)”

Changed as suggested (current lines 179-180).

17) “Line 169: remove the word “with.””

Changed as suggested (current line 183).

18) “Line 190: Move “only” to before “included” to improve flow.”

Changed as suggested (current lines 203-204).

19) “Line 237: Use “were” instead of “was””

To improve clarity, the sentence was reworded as follows (current lines 251-254): “Ninety-five percent (95%) confidence interval was calculated for each of the above mentioned indices of diagnostic test accuracy using a web-based application (MEDCALC - https://www.medcalc.org/calc/diagnostic_test.php).”

20) “Line 230: Use “pre-determined” instead of “pre-determine””

Changed as suggested (current lines 245-246).

21) “Line 312: Use “and indeed,””

Changed as suggested (current line 329). 

22) “Line 385: Use “Considering””

Changed as suggested (current lines 402).

23) “Line 426: Use “imbalance” instead of “unbalance””

Changed as suggested (current lines 439).

Additional typos evidenced during the review were corrected as follows:

- Former line 59 (current line 61): double space preceding “While” deleted.

- Former line 105 (current line 118): “University of Milano” changed to “University of Milan”

- Former line 153 (current line 167): comma between “i.e.” and “10x” deleted.

- Former line 287 (current line 304): comma between “represented” and “in” substituted by semicolon. 

- Former lines 340-342 (current lines 357-359): percentages were wrong, deriving from a former draft of the manuscript, including inconclusive cases and lacking reference to Table 3. The sentence now runs as follows: “Following the tabulation of cytological and histological diagnoses (S1 Table), 36 cases (46.15%) were classified as TP, 21 (26.92%) were TN, 20 (25.64%) were FN, and 1 (1.28%) was a FP (Table 3).”

- Former line 391 (current line 408): “of” added between “diagnosis” and “canine”.

- Former line 492 (current line 501): additional space preceding “Unfortunately” deleted.

---

## [Decision Letter · Decision Letter 1]

4 Oct 2019

PONE-D-19-20551R1

Evaluation of cytological diagnostic accuracy for canine splenic neoplasms: an investigation in 78 cases on adherence to STARD guidelines.

PLOS ONE

Dear Dr. Gambini,

Thank you for submitting your manuscript to PLOS ONE. After careful consideration, we feel that it has merit but does not fully meet PLOS ONE’s publication criteria as it currently stands. Therefore, we invite you to submit a revised version of the manuscript that addresses the points raised during the review process.

Please address the Reviewer's additional minor grammatical comments.

We would appreciate receiving your revised manuscript by Nov 18 2019 11:59PM. To enhance the reproducibility of your results, we recommend that if applicable you deposit your laboratory protocols in protocols.io, where a protocol can be assigned its own identifier (DOI) such that it can be cited independently in the future. For instructions see: http://journals.plos.org/plosone/s/submission-guidelines#loc-laboratory-protocols

We look forward to receiving your revised manuscript.

Kind regards,

Douglas H. Thamm, V.M.D.

Academic Editor

PLOS ONE

Reviewers' comments:

Reviewer's Responses to Questions

**Comments to the Author**

1. If the authors have adequately addressed your comments raised in a previous round of review and you feel that this manuscript is now acceptable for publication, you may indicate that here to bypass the “Comments to the Author” section, enter your conflict of interest statement in the “Confidential to Editor” section, and submit your "Accept" recommendation.

Reviewer #1: All comments have been addressed

Reviewer #2: (No Response)

2. Is the manuscript technically sound, and do the data support the conclusions?

Reviewer #1: Yes

Reviewer #2: Yes

3. Has the statistical analysis been performed appropriately and rigorously? 

Reviewer #1: Yes

Reviewer #2: Yes

4. Have the authors made all data underlying the findings in their manuscript fully available?

Reviewer #1: Yes

Reviewer #2: Yes

5. Is the manuscript presented in an intelligible fashion and written in standard English?

Reviewer #1: Yes

Reviewer #2: Yes

6. Review Comments to the Author

Reviewer #1: The manuscript is improved. No further comments. The manuscript is improved. No further comments. The manuscript is improved. No further comments. The manuscript is improved. No further comments.

Reviewer #2: Thank you for your thoughtful revision of this manuscript. The clarity of your study is markedly improved and I think the study is interesting and important.

I have some minor grammatical suggestions that I think will improve the readability of the manuscript, and there are two statements that I think could be clarified:

1. Remove "of" in line 57.

2. Lines 98-102 are not clear. Consider:

"Application of STARD guidlines in the current study allowed cross-tabulation of cytological results (i.e. the index test) against those of histopathology (i.e. the reference standard) to generate sensitivity and specificity data. These data have not been included in previous studies and will be useful for future researchers comparing diagnostic methods for canine splenic neoplasms."

3. Line 106, remove "a" before neoplasia (or say "a neoplasm")

4. Line 132, remove "considered as"

5. Lines 218, 220, 222: add "the" before each category name (the True Positive (TP), etc.)

6. Line 305, add "The" before "Time interval"

7. Line 326: Either add an "and" before therefore, or change the comma to a semicolon.

8. Line 332: a dash is missing in "non-neoplastic."

9. Line 368: change "ratio" to "ratios"

10. Line 444: This is not clear. Consider "Moreover, identification of vascular tumors (angiomas and HES) among false negative cases is not surprising since the architecture of these tumors often leads to significant peripheral blood contamination in aspirates."

11. Line 450: Change "by" to "of"

12: Line 451: Add "a" before low mitotic rate.

13: Line 462: Change the comma to a semicolon.

14: Line 471: Change "be considered also" to "also be considered."

15: Line 509: change "neoplasia, however high" to "neoplasia; however, high"

7. PLOS authors have the option to publish the peer review history of their article (what does this mean?). If published, this will include your full peer review and any attached files.

Reviewer #1: No

Reviewer #2: No

---

## [Author Response · Author response to Decision Letter 1]

22 Oct 2019

REBUTTAL LETTER – Manuscript PONE-D-19-20551

RESPONSES TO REVIEWER 2

1) “Remove "of" in line 57.”

Changed as suggested (current line 57).

2) “Lines 98-102 are not clear. Consider: "Application of STARD guidlines in the current study allowed cross-tabulation of cytological results (i.e. the index test) against those of histopathology (i.e. the reference standard) to generate sensitivity and specificity data. These data have not been included in previous studies and will be useful for future researchers comparing diagnostic methods for canine splenic neoplasms."”

Changed as suggested (current lines 98-103).

3) “Line 106, remove "a" before neoplasia (or say "a neoplasm")”

Changed as suggested (current line 107).

4) “Line 132, remove "considered as"”

Changed as suggested (current lines 132-133).

5) “Lines 218, 220, 222: add "the" before each category name (the True Positive (TP), etc.)”

Changed as suggested (current lines 219, 221, 223, 225).

6) “Line 305, add "The" before "Time interval"”

Changed as suggested (current line 306).

7) “Line 326: Either add an "and" before therefore, or change the comma to a semicolon.”

Changed as suggested (current line 327).

8) “Line 332: a dash is missing in "non-neoplastic."”

Changed as suggested (current line 333).

9) “Line 368: change "ratio" to "ratios"”

Changed as suggested (current line 369).

10) “Line 444: This is not clear. Consider "Moreover, identification of vascular tumors (angiomas and HES) among false negative cases is not surprising since the architecture of these tumors often leads to significant peripheral blood contamination in aspirates."”

Changed as suggested (current lines 445-447).

11) “Line 450: Change "by" to "of"”

Changed as suggested (current line 451).

12) “Line 451: Add "a" before low mitotic rate.”

Changed as suggested (current line 452).

13) “Line 462: Change the comma to a semicolon.”

Changed as suggested (current line 463).

14) “Line 471: Change "be considered also" to "also be considered."”

Changed as suggested (current line 472).

15) “Line 509: change "neoplasia, however high" to "neoplasia; however, high"”

Changed as suggested (current line 510).

Additional typos evidenced during the review were corrected as follows:

- Former line 45 (current line 45): double space preceding “Conversely” deleted.

- Former line 50 (current line 50): extra space preceding “Ultrasonographic” deleted.

- Former line 53 (current line 53): double space preceding “is” deleted.

- Former line 146 (current line 145): double space preceding “were” deleted.

- Former line 174 (current line 173): extra comma following “e.g.” deleted.

- Former line 182 (current line 181): double space preceding “specimens” deleted.

- Former line 190 (current line 189): extra space preceding “Neoplastic” deleted.

- Former line 347 (current line 346): double space preceding “Of” deleted.

- Former line 364 (current line 363): “was found, however severe marginal” reworded as “was found; however, severe marginal”.

- Former line 418 (current line 417): double space preceding “values” deleted.

- Former line 424 (current line 423): double space preceding “when” deleted.

- Former line 475 (current line 473): extra comma following “tumors” deleted.

- Former line 747 (current line 745): double space preceding “n/a” deleted.

---

## [Editor Report · Decision Letter 2]

25 Oct 2019

Evaluation of cytological diagnostic accuracy for canine splenic neoplasms: an investigation in 78 cases using STARD guidelines.

PONE-D-19-20551R2

Dear Dr. Gambini,

We are pleased to inform you that your manuscript has been judged scientifically suitable for publication and will be formally accepted for publication once it complies with all outstanding technical requirements.

With kind regards,

Douglas H. Thamm, V.M.D.

Academic Editor

PLOS ONE
---

## [Editor Report · Acceptance letter]

31 Oct 2019

PONE-D-19-20551R2 

Evaluation of cytological diagnostic accuracy for canine splenic neoplasms: an investigation in 78 cases using STARD guidelines. 

Dear Dr. Gambini:

I am pleased to inform you that your manuscript has been deemed suitable for publication in PLOS ONE. Congratulations! Your manuscript is now with our production department. 

With kind regards,

on behalf of

Dr. Douglas H. Thamm 

Academic Editor

PLOS ONE